# Systematic Review on Intra- and Extracochlear Electrical Stimulation for Tinnitus

**DOI:** 10.3390/brainsci11111394

**Published:** 2021-10-24

**Authors:** Kelly K. S. Assouly, Max J. Dullaart, Robert J. Stokroos, Bas van Dijk, Inge Stegeman, Adriana L. Smit

**Affiliations:** 1Department of Otorhinolaryngology and Head & Neck Surgery, University Medical Center Utrecht, 3584 Utrecht, The Netherlands; M.J.Dullaart-2@umcutrecht.nl (M.J.D.); R.J.Stokroos@umcutrecht.nl (R.J.S.); I.Stegeman@umcutrecht.nl (I.S.); A.L.Smit-9@umcutrecht.nl (A.L.S.); 2UMC Utrecht Brain Center, University Medical Center Utrecht, 3584 Utrecht, The Netherlands; 3Cochlear Technology Center, 2800 Mechelen, Belgium; BVanDijk@cochlear.com

**Keywords:** tinnitus, intracochlear electrical stimulation, extracochlear electrical stimulation, neuromodulation, systematic review

## Abstract

Several electrical stimulation patterns of the auditory nerve have been described for tinnitus relief, but there is no consensus on the most effective stimulation pattern. Therefore, we aim to systematically review the literature on the effect of intra- and extracochlear electrical stimulation patterns as a treatment option for patients with tinnitus. Only studies on intra- and extracochlear electrical stimulation for patients with tinnitus were included if the stimulation used did not concern standardized CI stimulation patterns to primarily rehabilitate hearing loss as intervention. A total of 34 studies met the inclusion criteria, with 10 studies (89 patients) on intracochlear electrical stimulation and 25 studies on extracochlear electrical stimulation (1109 patients). There was a high to medium risk of bias in 22 studies, especially due to lack of a non-exposed group and poor selection of the exposed group. All included studies showed subjective tinnitus improvement during or after electrical stimulation, using different stimulation patterns. Due to methodological limitations and low reporting quality of the included studies, the potential of intra- and extracochlear stimulation has not been fully explored. To draw conclusions on which stimulation patterns should be optimized for tinnitus relief, a deeper understanding of the mechanisms involved in tinnitus suppression is needed.

## 1. Introduction

Tinnitus is the perception of a sound without an external auditory input, often experienced as a ringing or buzzing sound in the ear or the head. Tinnitus can become severe and disabling, affecting quality of life and causing anxiety, depression and sleep disorders in those affected. The pathophysiology of tinnitus is still not fully understood. One main hypothesis is that tinnitus originates from maladaptive plasticity, causing an increase in spontaneous and synchronous activity in the auditory pathway [1]. So far, there is no curative treatment available, only tinnitus management therapies that reduce the burden.

A cochlear implant (CI) is an invasive device that transmits the external sound environment by electrically stimulating the auditory nerve of a deaf ear through the cochlea, thereby providing auditory sensation. Cochlear implantation aims to partially restore hearing and does not specifically target tinnitus [2]. In patients with severe to profound hearing loss, intracochlear electrical stimulation through CI showed positive effects on pre-operative tinnitus complaints, but tinnitus induction was reported in some cases; therefore, tinnitus reduction cannot be predicted yet [3,4,5,6]. It is still unclear how intra- and extracochlear electrical stimulation applied primarily for hearing improvement leads to tinnitus relief. Optimizing this electrical stimulation could lead to the development of a tinnitus-dedicated device and an efficient treatment for tinnitus relief, which might also be suitable for patients with less than severe hearing loss.

One of the challenges is to develop an electrical stimulation that evokes ‘silence’ instead of sound. Electrically stimulating neurons of the auditory nerve enables targeting the auditory pathway and thus may counteract tinnitus origins. This can be achieved using intracochlear stimulation by electrodes within the cochlea or, potentially, by extracochlear stimulation applied by electrodes outside the cochlea. In these situations, the major issue now is to identify electrical patterns that induce suitable and substantial tinnitus relief.

So far, several electrical stimulation patterns of the auditory nerve have been described for tinnitus relief, but no consensus on the most effective type of stimulus exists [7,8]. Therefore, in this paper, we aim to systematically review the literature on the effectiveness of intra- and extracochlear electrical stimulation techniques and patterns as a treatment option for patients with tinnitus.

## 2. Materials and Methods

### 2.1. Protocol and Registration

The protocol of this systematic review can be found in PROSPERO with registration number CRD42020180652. We followed the Preferred Reporting Items for Systematic Reviews and Meta-Analyses (PRISMA) format for this systematic review [9].

### 2.2. Search Strategy

We conducted a systematic search in PubMed, Embase, the Cochrane Library, CINAHL, and Web of Science. Search terms and their synonyms of domain (tinnitus) and determinant (electrical stimulation) were used in title, abstract and medical subject headings (MeSH) terms. The search syntaxes can be found in Table 1. In addition to electronic database searches, reference lists were scanned to identify additional studies. We searched in ClinicalTrials.gov for ongoing trials and protocols. The search was conducted on 7 August 2021.

### 2.3. Eligibility Criteria

We defined our research question and selected eligibility criteria (Table 2) based on the Participants, Intervention, Comparators and Outcomes (PICO) design [10]. Articles published or accepted for publication in peer-reviewed academic journals and ongoing trials were eligible for screening without publication date restriction. Studies on intra-and extracochlear electrical stimulation for patients with tinnitus were included only if the stimulation used did not concern standardized CI stimulation patterns to primarily rehabilitate hearing loss as intervention.

We excluded studies with a non-original study design, animal studies or studies without an available abstract after the title/abstract screening. Exclusion criteria were studies without an available full text or studies presenting overlapping populations. We contacted corresponding authors to retrieve full text articles if these were not available in our databases or for clarification and further data. In the case of overlap, the most complete publication was included.

### 2.4. Study Selection

After removal of duplicates, two authors (K.K.S.A. and M.J.D.) independently performed the title/abstract and full text screening of the retrieved studies, according to our inclusion and exclusion criteria (Table 2). The screening tool used was Rayyan [11]. Conflicts about selection were resolved through discussion with two additional reviewers (A.L.S. and I.S.).

### 2.5. Data Collection and Analysis

#### 2.5.1. Quality Assessment of the Studies

Two authors (K.K.S.A. and M.J.D.) independently assessed the risk of bias (RoB). We used the Newcastle–Ottawa quality assessment Scale (NOS) to evaluate risk of bias in cohort studies [12]. The NOS uses three domains to evaluate risk of bias: selection, comparability and exposure for case-control studies and selection, comparability and outcomes for cohort studies (NOS checklist available in Table A1). Items were scored using stars. An overall risk of bias judgment was determined based on the total score: ● high risk of bias (0–3), ◐ medium risk of bias (4–6), ○ low risk of bias (7–9). Any discrepancies were resolved through discussion between the two reviewers and then by consulting with two additional reviewers (A.L.S. and I.S.). Case reports and ongoing trials were not applicable for quality assessment.

#### 2.5.2. Data Extraction and Synthesis

All study characteristics and outcomes were extracted independently and then compared by two authors (K.K.S.A. and M.J.D.). The following information was extracted: study characteristics (first author, publication year and study design), patient characteristics (number of patients, age, gender and inclusion criteria), intervention parameters (stimulation location, stimulation mode, stimulation intensity, pulse rate, polarity and, if available, duration of treatment), tinnitus outcomes, follow-up and adverse effects. We presented outcomes separately by type of stimulation (intracochlear or extracochlear). When the data were incomplete or only reported on a graph, we contacted the corresponding authors for details. If available, outcomes were reported with their corresponding 95% confidence intervals (95% CI) or the standard deviation (SD), and *p*-value. The *p*-value is the result of a statistical comparison test between the tinnitus questionnaire scores used at different follow-up period or groups (Table 5) or for specific parameter values (Table 6). The cut-off of the p-value used to indicate a statistically significant result was established as described in the corresponding studies. We did not perform statistical analysis on the extracted data.

Because of the heterogeneity of the studies in methods, inclusion of participants, interventions and assessment of outcomes, we did not conduct a meta-analysis but instead performed a descriptive synthesis of the results.

#### 2.5.3. Outcome Measures

The primary outcome of this review is the self-reported experience of tinnitus of specific electrical stimulation parameters, in which tinnitus was measured by general questions or validated questionnaires assessing one or more aspects of the tinnitus (e.g., loudness, severity, distress, annoyance, irritability, awareness, or intrusiveness).

Secondary outcomes were adverse effects. We considered negative effects related to electrode placement or electrical stimulation (e.g., infection, pain or discomfort, facial nerve palsy, dizziness, tinnitus increase) as relevant harms.

#### 2.5.4. Tinnitus Outcomes

The tinnitus questionnaires used are the Tinnitus Functional Index (TFI), Tinnitus Handicap Inventory (THI), Tinnitus Handicap Questionnaire (THQ), Tinnitus Questionnaire (TQ) and the Visual Analogue Scale (VAS) for tinnitus experience.

The TFI contains 25 questions about eight domains: intrusiveness, sense of control, cognitive interference, sleep disturbance, auditory difficulties attributed to tinnitus, relaxation, quality of life and emotional distress. Possible answers are rated on a scale of 0 to 10, or 0% to 100%. An overall TFI score of 0 to 100 can be calculated, where a total score of more than 53 indicates severe tinnitus burden. A clinically relevant reduction is characterized by a decrease of 13 points or more [13].

The THI questionnaire contains 25 questions characterizing the effect of tinnitus on a patient’s emotions and daily life. Possible answers are ‘no’ (0 points), ‘sometimes’ (2 points) and ‘yes’ (4 points). An overall THI score can be calculated, resulting in five different tinnitus grades: no or slight handicap (0–16 points), mild handicap (18–36 points), moderate handicap (38–56 points), severe handicap (58–76 points) and catastrophic handicap (78–100 points) [14]. A decrease of seven points or more can be interpreted as a clinically relevant reduction of the tinnitus burden [15].

The THQ assesses the handicapping effect of tinnitus with 27 questions organized in a three-factor structure [16]. The three factors reflect the physical, emotional, and social consequences of tinnitus (Factor 1), hearing ability of the patient (Factor 2), and the patient’s view of tinnitus (Factor 3). Each question can be scored on a scale from 0 to 100, providing a total score also ranging from 0 to 100. This questionnaire can be used to compare the patient’s tinnitus handicap and to monitor progress with treatment.

The TQ measures distress caused by tinnitus with 52 questions divided into six domains: emotional and cognitive distress, intrusiveness, auditory perceptual difficulties, sleep disturbances and associated somatic complaints [17]. Three answers are possible for every question: ‘true’ (0 point), ‘partly true’ (1 point) or ‘not true’ (2 points). A total TQ score from 0 to 84 points can be reached, where a higher score indicates more distressing tinnitus. A clinically relevant reduction is characterized by a decrease of 12 points or more [18].

Single-item questionnaires based on a visual analogue scale (VAS) can be used to assess only one characteristic of tinnitus: loudness (VAS-L), severity, distress, annoyance (VAS-A), irritability, awareness and intrusiveness. The VAS consists of a horizontal or vertical line anchored at both ends by a verbal descriptor referring to the tinnitus characteristics. The tinnitus characteristic is scored from 0 (not at all) to 10 or to 100 (extremely). A single question asks the patient to tick the line on the point that best matches to his or her tinnitus characteristic.

In the case of self-reported testimony, total tinnitus suppression is defined as suppression of the tinnitus percept as long as the electrical stimulation is applied.

#### 2.5.5. Electrical Stimulation Parameters

We assessed the stimulation parameters used for intra- or extracochlear stimulation. We extracted four main parameters characterizing stimulation patterns: electrode location (E), current level (C), pulse rate (PR) and polarity (P) (Table 5). Intracochlear stimulation was always provided through a CI. Extracochlear stimulation was grouped into three different sites in the inner ear: promontory, oval window or round window. By convention, intracochlear electrical stimulation is characterized by an alternating current with charge-balanced biphasic pulse trains. Extracochlear stimulation can be delivered through direct (DC) or alternating current (AC) depending on the device used. For both modes, the current level, measured in amperes (A), and polarity, anodic or cathodic, can be adjusted to provide specific stimulation patterns and were reported. The pulse rate, measured in Hertz (Hz) or pulse per second (pps), is only relevant in AC mode. Occasionally, amplitude modulation can be performed, using a carrier wave to obtain specific patterns. The carrier wave and its specificity were reported, if applicable.

## 3. Results

### 3.1. Search Strategy and Study Selection

The selection process is summarized in the PRISMA flowchart in Figure 1. The search resulted in 7101 articles after removal of duplicates. After title and abstract screening, 69 articles remained for full text screening.

After full-text screening, 36 articles were excluded. Fifteen studies did not report on intra- or extracochlear electrical stimulation for tinnitus [19,20,21,22,23,24,25,26,27,28,29,30,31,32]. Three studies reported only on standard CI stimulation patterns to rehabilitate hearing loss [33,34,35]. Due to lack of response from the corresponding authors contacted, full text was not available for 11 studies [36,37,38,39,40,41,42,43,44,45,46]. One study did not have an original design [47]. One publication was only available in Japanese, and the two screeners were not able to have it translated [48]. We found overlapping populations in five studies. Four studies reported tinnitus outcomes of the same population [49,50,51,52]. We included the publication with the most complete data [51]. This also applies for the overlapping studies by Matsushima et al.; therefore, we excluded two studies [53,54].

Finally, 33 studies were selected for further analysis and data extraction [51,55,56,57,58,59,60,61,62,63,64,65,66,67,68,69,70,71,72,73,74,75,76,77,78,79,80,81,82,83,84,85,86,87]. Of these, 26 were prospective cohort studies. There were four case series [55,67,69,79], two case reports [70,71], and one pilot study [64]. Additionally, one ongoing study at the Mayo clinic, investigating the effect of promontory electrical stimulation, was included in our selection [84].

### 3.2. Quality of the Included Studies

We assessed the quality of the included studies, using the NOS tool. The results of our critical appraisal can be found in Table 3. There were nine (29%) studies that had a low risk of bias [51,56,58,61,62,65,72,82,83]. All these studies selected homogeneous populations, using inclusion criteria based on tinnitus severity and hearing loss, which led to higher quality. Studies with the highest score had a non-exposed group to compare outcomes with the intervention group [51,56,59,61,62,65]. Eleven (35.5%) studies had a moderate risk of bias, in which neither tinnitus outcomes nor self-report tinnitus experience were available before stimulation [55,57,59,60,63,67,69,76,79,85,86]. The overall risk of bias was considered high in 11 (35.5%) studies [64,66,68,73,74,75,77,78,80,81,87]. This was due to lack of a non-exposed group and poor representativeness of the exposed group. A poor representativeness corresponded to a selection of individuals or to a lack of description of the study population. These 11 studies did not report on pre-stimulation tinnitus outcome, nor on self-report tinnitus experience.

### 3.3. Data Extraction of Study Characteristics

We contacted eight authors for additional data [57,65,67,69,72,82,83], of which six responded to our request [57,67,69,72,82,83].

#### 3.3.1. Study Population

The characteristics of studies investigating the effect of intracochlear electrical stimulation on tinnitus can be found in Table 4, and studies assessing extracochlear electrical stimulation are presented in Table 4b. In total, 89 tinnitus patients were treated with intracochlear electrical stimulation, and 1109 with extracochlear stimulation. The sample sizes varied between different study designs, from 1 individual in a case report [70,71] to 168 patients in a parallel group design [59]. Tinnitus severity was not used as a selection criterion in all studies. Among the studies assessing intracochlear stimulation, all patients were implanted for sensorineural hearing loss, except for the study by Olze et al. in which this information was not available [64]. In studies using extracochlear stimulation, the hearing profiles were more diverse, ranging from normal hearing [61,62,68,86] to profound sensorineural hearing loss [58,69,74,75,77,78,87].

#### 3.3.2. Intervention

Nine studies (26%) investigated the effect of intracochlear electrical stimulation on tinnitus [51,57,64,65,70,71,72,82,83]. One study (3%) evaluated both types of stimulation: intracochlear stimulation in 3 patients and extracochlear stimulation in 11 patients [67]. Lastly, twenty-four (71%) studies assessed extracochlear electrical stimulation and its effect on tinnitus burden. Fifteen studies performed promontory stimulation [55,56,58,59,60,61,62,63,66,68,74,84,85,86,87], three studies used round window stimulation [69,76,78] and one tested oval window stimulation [79]. Five studies reported the outcomes of promontory and round window stimulation [73,75,77,80,81].

Among the studies assessing intracochlear electrical stimulation, eight performed acute stimulation [57,64,67,70,71,72,82,83], ranging between 500 milliseconds to 15 min, and two performed chronic stimulation [51,65]. Seven studies assessing extracochlear stimulation performed chronic stimulation [63,69,73,76,78,79,84]. The follow-up with outcome assessment varied between a few minutes after stimulation for punctual stimulation at the clinic to 3.5 years after placement and activation of a round window implant [69].

**Table 3 brainsci-11-01394-t003:** Quality assessment of the included studies based on the NOS.

Study (Author, Year)	Study Design	NOS Tool									
		Selection				Comparability	Outcome			Total	Risk of Bias
		(1)	(2)	(3)	(4)	(1)	(1)	(2)	(3)		
Aran et al., 1981 [77]	Cohort	☆	NA	☆	☆	☆★	☆	★	☆	2	●
Arts et al., 2015 [82]	Cohort	★	NA	☆	★	★★	★	★	★	7	○
Arts et al., 2016 [51]	Cohort	★	★	☆	★	★★	★	★	★	8	○
Cazals et al., 1978 [75]	Cohort	☆	NA	☆	☆	☆★	☆	★	★	3	●
Cazals et al., 1984 [78]	Cohort	☆	NA	☆	☆	☆★	☆	★	★	3	●
Chang et al., 2012 [83]	Cohort	★	NA	☆	★	★★	★	★	★	7	○
Daneshi et al., 2005 [56]	Cohort	★	★	☆	★	★★	★	★	★	8	○
Dauman et al., 1993 [57]	Cohort	☆	NA	☆	★	★★	★	★	★	6	◐
Di Nardo et al., 2009 [58]	Cohort	★	NA	☆	★	★★	★	★	★	7	○
Graham et al., 1977 [74]	Cohort	☆	NA	☆	☆	☆☆	☆	★	☆	1	●
Hazell et al., 1993 [76]	Cohort	★	NA	☆	☆	★★	☆	★	★	5	◐
House et al., 1984 [73]	Cohort	☆	NA	☆	☆	★☆	☆	★	☆	2	●
Ito et al., 1994 [87]	Cohort	☆	NA	☆	☆	☆☆	☆	★	★	2	●
Kloostra et al., 2020 [72]	Cohort	★	NA	☆	★	★★	★	★	★	7	○
Konopka et al., 2001 [60]	Cohort	★	☆	☆	☆	★★	★	★	★	6	◐
Konopka et al., 2008 [59]	Cohort	★	★	☆	☆	☆★	☆	★	★	5	◐
Mahmoudian et al., 2013 [62]	Cohort	★	★	☆	★	★★	★	★	★	8	○
Mahmoudian et al., 2015 [61]	Cohort	★	★	☆	★	★★	★	★	★	8	○
Matsushima et al., 1994 [85]	Cohort	★	NA	☆	☆	★☆	☆	★	★	4	◐
Matsushima et al., 1996a [63]	Cohort	☆	NA	☆	☆	★★	☆	★	★	4	◐
Matsushima et al., 1996b [55]	Cohort	★	NA	☆	☆	★☆	☆	★	★	4	◐
Okusa et al., 1993 [86]	Cohort	★	NA	☆	☆	☆☆	★	★	★	4	◐
Olze et al., 2018 [64]	Cohort	☆	NA	☆	☆	☆☆	★	★	★	3	●
Péan et al., 2010 [79]	Cohort	★	NA	☆	☆	★★	★	★	★	6	◐
Portmann et al., 1979 [80]	Cohort	★	NA	☆	☆	☆☆	☆	★	★	3	●
Portmann et al., 1983 [81]	Cohort	☆	NA	☆	☆	☆☆	☆	★	★	2	●
Punte et al., 2013 [65]	Cohort	★	★	☆	★	★★	★	★	★	8	○
Rothera et al., 1986 [66]	Cohort	☆	NA	☆	☆	☆☆	☆	★	★	2	●
Rubinstein et al., 2003 [67]	Cohort	★	☆	☆	☆	★★	★	★	★	6	◐
Watanabe et al., 1997 [68]	Cohort	★	NA	☆	☆	☆☆	☆	★	★	3	●
Wenzel et al., 2015 [69]	Cohort	★	NA	☆	☆	★★	★	★	★	6	◐
Rothholtz et al., 2009 [71]	Case report	NA	NA	NA	NA	NA	NA	NA	NA	NA	NA
Zeng et al., 2011 [70]	Case report	NA	NA	NA	NA	NA	NA	NA	NA	NA	NA
Carlson et al., 2020 [84]	Cohort	NA	NA	NA	NA	NA	NA	NA	NA	NA	NA

★ 1 point; ☆ 0 point; ● High risk of bias (0–3); ◐ Medium risk of bias (4–6); ○ Low risk of bias (7–9); NA: not applicable.

**Table 4 brainsci-11-01394-t004:** Study characteristics of the included studies. (**a**) Studies reporting on intracochlear electrical stimulation; (**b**) Studies reporting on extracochlear electrical stimulation.

(a) Studies Reporting on Intracochlear Electrical Stimulation										
Authors, Year	Study Design	N (Tinnitus Patients)	Demographics		Study Population		Stimulation Type	Outcomes		
			Gender M (F)	Age (SD/Range)	Tinnitus Criteria	Hearing Loss		Follow-Up (Max)	Tinnitus Question/ Questionnaire	Harms Reported
Arts et al., 2015 [82]	PCS	11	6 (5)	60.1 (6.4)	VAS-L > 2, THI > 16	Severe to profound SNHL	CI	DS	THI, VAS-L	None
Arts et al., 2016 [51]	PCS	10	5 (5)	48.2 (12.5)	VAS-L > 7, THI > 38, TQ > 42	SSD	CI	3 months	THI, TQ, VAS-L	None
Chang et al., 2012 [83]	PCS	13	2 (11)	60.8 (13.6)	NI	Severe to profound SNHL	CI	DS	THI, TSI	NI
Dauman et al., 1993 [57]	PCS	2	NI	38–51 *	Bilateral	Profound SNHL	CI	DS	THQ, VAS-L	NI
Kloostra et al., 2020 [72]	PCS	19	12 (7)	60.6 (43–78)	Chronic, constant	Bilateral severe SNHL	CI	AS	THI, THQ, VAS-L	NI
Olze et al., 2018 [64]	Pilot	6 (4)	NI	NI	NI	NI	CI	AS	VAS-L	Yes
Punte et al., 2013 [65]	PCS	14	5 (9)	NI	VAS-L ≥ 6	Profound SNHL	CI	6 months	TQ, VAS-L	NI
Rothholtz et al., 2009 [71]	Case report	1	1 (0)	NI	Unilateral, debilitating	SSD	CI	AS	VAS-L	NI
Rubinstein et al., 2003 [67]	Case series	14	NI	NI	Bothersome	Severe to profound HF SNHL	CI (3), RW (11)	3 days	THQ, VAS-L, VAS-A	Yes
Zeng et al., 2011 [70]	Case report	1	1 (0)	46	NI	Profound SNHL	CI	AS	VAS-L	Yes
**(b) Studies Reporting on Extracochlear Electrical Stimulation**										
**Authors, Year**	**Study** **Design**	**N (Tinnitus Patients)**	**Demographics**		**Study Population**			**Outcomes**		**Harms** **Reported**
			**Gender M (F)**	**Age (SD/Range)**	**Tinnitus** **Criteria**	**Hearing Loss**	**Stimulation Type (OW, PM, RW)**	**Follow-Up (Max)**	**Tinnitus Question/** **Questionnaire**	
Aran et al., 1981 [77]	PCS	106 (84)	NI	NI	NI	Profound SNHL	RW, PM	DS	Self-report	Yes
Cazals et al., 1978 [75]	PCS	16 (6)	11 (5)	NI	NI	Severe to profound HL	RW (13), PM (3)	AS	Self-report	Yes
Cazals et al., 1984 [78]	PCS	4 (1)	1	NI	NI	Totally deaf	RW	3 months	Self-report	Yes
Daneshi et al., 2005 [56]	PCS	52	32 (20)	42.2 (21–67) (PM)	Moderate to severe	Moderate to severe HL (PM)	PM (32), CI (20)	50 days	TQ, TSS	NI
Di Nardo et al., 2009 [58]	PCS	11	4 (7)	34–64	Severe	Profound SNHL	PM, CI (control)	1 month	THI	NI
Graham et al., 1977 [74]	PCS	13 (9)	NI	NI	NI	Profound SNHL	PM	AS	Self-report	Yes
Hazell et al., 1993 [76]	PCS	9	NI	NI	Severe	Unilateral deafness	RW	NI	Self-report	NI
House et al., 1984 [73]	PCS	130 (125)	NI	NI	NI	HL in varying degrees	PM, RW	1 week	Self-report	NI
Ito et al., 1994 [87]	PCS	40 (30)	18 (12)	46.6 (18–63)	NI	Severe HL or totally deaf	PM	AS	Self-report	NI
Konopka et al., 2001 [60]	PCS	111	91 (20)	55.5 (15–67)	NI	NIHL and SNHL	PM	3 months	VAS-L	Yes
Konopka et al., 2008 [59]	PCS	248 (168)	NI	23–78	NI	NIHL and SNHL	PM	1 month	Assessment of subjective feelings	Yes
Mahmoudian et al., 2013 [62]	PCS	44	32 (12)	44.71 (18–65)	TQ > 44 VAS-L > 6	PTA(HF) < 60 dB	PM	1 week	VAS-L	NI
Mahmoudian et al., 2015 [61]	PCS	28	18 (10)	35.33 (22–45)	TQ > 44, THI > 39, VAS-L > 6	Normal hearing	PM	AS	TQ, VAS-L	NI
Matsushima et al., 1994 [85]	PCS	112	76 (36)	53 (19–73)	NI	NI	PM	1 month	Interview	NI
Matsushima et al., 1996a [63]	Case series	4	2 (2)	51.8 (44–57)	NI	HL in varying degrees	PM	3 months	Self-report	Yes
Matsushima et al., 1996b [55]	PCS	47	24 (23)	60.4 (42–75)	NI	HL in varying degrees	PM	DS	Self-report	NI
Okusa et al., 1993 [86]	PCS	65	NI	47 (17–72)	NI	Normal to profound SNHL	PM	>3 days	VAS-L	Yes
Péan et al., 2010 [79]	Case series	4	NI	NI	Severe	Unilateral deafness	OW	121 days	DET	NI
Portmann et al., 1979 [80]	PCS	28 (15)	NI	NI	NI	NI	RW (11), PM (7)	DS	Self-report	NI
Portmann et al., 1983 [81]	PCS	120 (72)	NI	NI	NI	NI	RW, PM	few days	Self-report	NI
Rothera et al., 1986 [66]	PCS	20 (16 ears)	NI	NI	NI	NI	PM	AS	Self-report	NI
Rubinstein et al., 2003 [67]	Case series	14	NI	NI	Bothersome	Mild to moderate SNHL	CI (3), RW (11)	3 days	THQ, VAS-L, VAS-A	Yes
Watanabe et al., 1997 [68]	PCS	56	35 (21)	49.4 (21–71)	NI	With and without HL	PM	1 month	Self-report	Yes
Wenzel et al., 2015 [69]	Case series	3	2 (1)	43.3 (38–50)	Unilateral, resistant to pharmacological treatment	Unilateral severe to profound SNHL	RW	3.5 years	THI, VAS-L, VAS-A	Yes
Carlson et al., 2020 [84]	on going	21	NI	NI	TFI > 52, THI > 56, VAS-L > 5	Normal to moderate SNHL	PM	1 week	THI, TFI, VAS-P	NI

AS: after stimulation; CI: cochlear implant; DET: distress evaluation tinnitus; DS: during stimulation; HL: hearing loss; HF: high frequencies; LF: low frequency; N: number of patients; NI: no information; OW: oval window; PCS: prospective cohort study; PM: promontory; PTA: pure tone average; RW: round window; SNHL: sensorineural hearing loss; SD: standard deviation; SSD: single-sided deafness; THI: tinnitus handicap questionnaire; TQ: tinnitus questionnaire; TSS: tinnitus severity scale; VAS-A: visual analogue scale annoyance; VAS-L: visual analogue scale loudness. * Extracted from a graph. In the studies of Aran et al. (1981) [77] and Cazals et al. (1978) [75], tinnitus was assessed by asking patients to raise hand and describe the sensation when they experienced a change during stimulation. Other studies using self-report as a tinnitus outcome did not provide further details on the instructions given to patients.

#### 3.3.3. Outcomes

Twelve studies reported on tinnitus distress or burden using multi-item questionnaires: one study used the TFI [84], seven studies the THI [51,58,69,72,82,83,84], three studies the THQ [57,67,72] and four studies the TQ [51,56,61,65]. The used single-item questionnaires assessed tinnitus loudness (VAS-L) in 14 studies [51,57,60,61,62,64,65,67,69,70,71,72,82,86], annoyance (VAS-A) in two studies [67,69] and pain (VAS-P) in one study [84]. Among the studies using tinnitus questionnaires, seven used only one specific tinnitus questionnaire [58,60,62,64,71,79,86], where others used two or more. Tinnitus matching was performed in 14 studies [51,56,58,59,60,61,62,65,67,68,74,79,82,86].

### 3.4. Synthesis of Results

#### 3.4.1. Tinnitus Outcomes

Of the 34 studies included, 10 reported scores from tinnitus questionnaires pre- and post-stimulation. Seven out of ten studies performed statistical analyses. A summary of the effects is detailed in Table 5.

##### THI

Arts et al. (*n* = 10) found no statistically significant difference in THI between personalized stimulation through CI (a combination of stimulation parameters chosen for each patient) and standard stimulation through CI (stimulation dependent of the environmental sound defined by an audiologist for speech perception purposes) after one and three months of stimulation, respectively (THI after three months of standard stimulation: 31.0 (IQR: 22.0–46.5), THI after three months of personalized stimulation: 40.0 (IQR: 25.0–52.0), *p* = 0.15) [51]. 

Di Nardo et al. (*n* = 11) showed a decrease in THI after promontory stimulation compared to pre-stimulation but did not report a *p*-value (THI pre-stimulation: 49.1 (SD: 22.9); THI post-stimulation: 33.6 (SD: 26.0)) [58]. The three case series of Wenzel et al. showed a decrease in THI from activation of the round window implant to nine months after implantation, but did not report the statistical significance of the outcome (THI activation: 83.33 (SD: 11.85), THI 9 months: 78.33 (SD: 25.66)) [69].

##### TQ

Arts et al. (*n* = 10) found no statistically significant difference in TQ between personalized stimulation and standard stimulation after one and three months of stimulation through CI respectively (TQ after three months standard stimulation: 23.5 (IQR: 13.75–43.25), TQ after three months of personalized stimulation: 30.0 (IQR: 22.5–34.75), *p* = 0.18) [51].

Daneshi et al. (*n* = 32) showed a statistically significant decrease in TQ between pre- and post-promontory stimulation (TQ pre-stimulation: 50.66 (SD: 19.34), TQ post-stimulation: 39.03 (SD: 20.35), *p* = 0.001) [56].

**Table 5 brainsci-11-01394-t005:** Extracted data of tinnitus distress outcomes. (**a**) Studies reporting on intracochlear electrical stimulation; (**b**) Studies reporting on extracochlear electrical stimulation.

(a) Studies Reporting on Intracochlear Electrical Stimulation										
Authors, Year	N	ES Configuration		Outcomes						
				Questionnaire	Group (CI, Control)	BS	DS	AS	FU	*p*-Value
Arts et al., 2015 [82]	11	C: −10%		% VAS-L reduction	CI	0	7 (−4.5–29)°	NI	DS	>0.05
		C: 10% DR		% VAS-L reduction		0	18 (−2.25–76)°	NI	DS	>0.05
		C: 50% DR		% VAS-L reduction		0	22.5 (9.5–87.75)°	NI	DS	0.033
		C: 80% DR		% VAS-L reduction		0	56.5 (−3.5–94)°	NI	DS	0.014
		E: basal (x1, x3)		% VAS-L reduction		0	15 (4.5–29.5)°	NI	DS	>0.05
		E: central (x1, x3)		% VAS-L reduction		0	25 (5–60)°	NI	DS	>0.05
		E: apical (x1, x3)		% VAS-L reduction		0	4.5 (−6.5–39.5)°	NI	DS	>0.05
		E: pitch-matched (x1, x3)		% VAS-L reduction		0	22.5 (2–28.5)°	NI	DS	>0.05
Arts et al., 2016 [51]	10	combinations of E, C, PR, P, A, pulse width		THI	CI	45 (40–53)°	40.00 (25.00–44.50)	NI	1 month	0.06¨
							40.00 (25.00–52.00)	NI	3 months	0.15¨
		standard			CI (control)	45 (40–53)°	38.00 (21.50–44.50)	NI	1 month	0.06¨
							31.00 (22.00–46.50)	NI	3 months	0.15¨
		combinations of E, C, PR, P, A, pulse width		VAS-L	CI	7.1 (6.4–7.7)°	3.35 (2.68–6.95)	NI	1 month	0.25¨
							3.40 (2.40–7.63)	NI	3 months	0.39¨
		standard			CI (control)	7.1 (6.4–7.7)°	3.15 (2.00–5.80)	NI	1 month	0.25¨
							3.50 (1.55–6.63)	NI	3 months	0.39¨
		combinations of E, C, PR, P, A, pulse width		TQ	CI	40 (33–51)°	30.00 (19.25–38.25)	NI	1 month	0.77¨
							30.00 (22.50–34.75)	NI	3 months	0.18¨
		standard			CI (control)	40 (33–51)°	27.00 (23.50–38.50)	NI	1 month	0.77¨
							23.50 (13.75–43.25)	NI	3 months	0.18¨
Chang et al., 2012 [83]	13	combinations of E, C, PR		THI	CI	26.8 (17.6)	NI	NI	DS	NI
Kloostra et al., 2020 [72]	19	combinations of E, C, PR		THI	CI	28.4 (22.9)	NI	NI	NA	NI
				THQ	CI	38.6 (27.2)	NI	NI	NA	NI
Punte et al., 2013 [65]	7 (7 control)	E: 1 most basal		VAS-L	CI	8.3 (1.1)	7.2 *	7.9 *	pre	
		E: 2 most basal					7.3 *	7.2 *	1 week	>0.05
		E: 3 most basal					7.0 *	7.1 *	1 week	>0.05
		E: 4 most basal					7.0 *	7.5 *	1 week	>0.05
		E: all					4.4 (1.3)	7.5 *	1 week	0.027
							3.5 (1.7)	8.1 *	6 months	0.042
					Control (no CI)	8.8 (1.0)	NI	8.7 (0.8)	6 months	>0.05
		E: all		TQ	CI	60 (15.6)	NI	39.4 (12.4)	6 months	<0.05
					Control (no CI)	58.9 (27.4)	NI	56.3 (25.4)	6 months	>0.05
Rothholtz et al., 2019 [71]	1	E: E2 C: 120 μA PR: 60 pps		VAS-L	CI	5	0	6	200 ms AS	NI
Zeng et al., 2011 [70]	1	E: apical, C: 100 mA, PR: 100 Hz, bi-phasic (107.8 ms/phase), loudness 6		VAS-L	CI	4 *	0 *	7 *	~100 ms AS	NI
		1: apical, 2: 100 mA, 3: 100 Hz, bi-phasic (107.8 ms/phase), loudness 3,				4 *	0 *	5.5 *	~100 ms AS	NI
		1: basal, 2: 100 mA, 3: 100-Hz, bi-phasic (107.8 ms/phase), loudness 6				4 *	5 *	6 *	~300 ms AS	NI
		1: basal, 2: 150 mA, 3: 5000 Hz, bi-phasic (32.3 ms/phase), loudness 5				4 *	5 *	5 *	~100 ms AS	NI
**(b) Studies Reporting on Extracochlear Electrical Stimulation**										
**Authors, Year**	**N**	**ES Configurations**		**Outcomes**						
		**AC/DC**	**Parameter(s) Tested**	**Questionnaire**	**Group (CI, Control, PM, RW)**	**BS**	**DS**	**AS**	**FU**	* **p** * **-Value**
Daneshi et al., 2005 [56]	32	AC	C: 60–500 μA PR: 50–600Hz	TQ	PM	50.66 (19.34)	NI	39.03 (20.35)	50 days	0.001 0.49¨
	20	AC	standard		CI (control)	52.84 (14.52)	NI	38.45 (13.99)	50 days	0.001 0.49¨
Di Nardo et al., 2009 [58]	11	DC+	C: 0–500 μA PR: 50–1600 Hz	THI	PM	49.1(22.9)	NI	33.6 (26.0)	1 month	NI
Mahmoudian et al., 2013 [62]	44	AC	C: 60–500 μA	VAS-L	PM (RI)	6.83 (1.37)	NI	3.13 (1.65)	AS	<0.05 <0.05¨ (NRI) <0.05¨ (placebo)
					PM (NRI)	6.90 (1.17)	NI	6.65 (1.04)	AS	>0.05 <0.05¨ (RI)>0.05¨ (placebo)
					placebo (control)	6.86 (1.27)	NI	6.68 (1.25)	AS	>0.05 <0.05¨ (RI) >0.05¨ (NRI)
Mahmoudian et al., 2015 [61]	28	AC	C: 50–500 μA	VAS-L	PM (RI)	6.38 (1.26)	NI	2.92 (1.75)	AS	<0.05 <0.05¨ (NRI) <0.05¨ (placebo)
					PM (NRI)	7.00 (1.25)	NI	6.73 (1.03)	AS	>0.05 <0.05¨ (RI) >0.05¨ (placebo)
					placebo (control)	7.00 (1.21)	NI	6.75 (1.23)	AS	>0.05 <0.05¨ (RI) >0.05¨ (NRI)
Wenzel et al., 2015 [69]	3	AC	C, PR, pulse duration	THI	RW	NI	83.33 (11.85) *	78.33 (25.66) *	9 months	NI
				VAS-L	RW	NI	8.0 (2.65) *	6.33 (5.51) *	9 months	NI
				VAS-A	RW	NI	8.67 (1.53) *	6.33 (5.51) *	9 months	NI

A: amplitude modulation; AS: after stimulation; BS: before stimulation; C: current level; CI: cochlear implant; DS: during stimulation; E: electrode location; FU: follow-up period; NA: not applicable; NI: no information; NRI: non-residual inhibition group; P: polarity; PM: promontory; PR: pulse rate; RI: residual inhibition group; RW: round window; * extracted from a graph; ° extracted from raw data not available in the publication. The *p*-value is the results of a comparison test between the two scores of the same line. It refers either to pre-, intra-, or post-stimulation scores, or to intra- and post-stimulation scores, except for *p*-value with ¨, which is the result from a comparison between the intervention group and the control group. Significant *p*-values are in bold. In the study of Punte et al. (2013), no comparison between groups was performed [65]. In the study of Wenzel et al. (2005), the follow-up period was restricted to 6 months because one patient received a speech coding program at 6 months post-implantation, which is out of the scope of this review [69].

##### VAS-L

Arts et al. (*n* = 10) found no statistically significant difference in VAS-L between personalized stimulation and standard stimulation after one and three months of stimulation through CI respectively (VAS-L after 3 months standard stimulation: 3.5 (IQR: 1.55–6.63), VAS-L after three months of personalized stimulation: 3.4 (IQR: 2.4–7.63), *p* = 0.039) [51]. In another study, Arts et al. (*n* = 11) showed a statistically significant decrease in VAS-L measured before and during stimulation with suprathreshold stimulation (50% dynamic range: 22.5 (IQR: 9.5–87.75) % VAS-L reduction, *p* = 0.033; 80% dynamic range: 56.5 (IQR: −3.5–94.0) % VAS-L reduction, *p* = 0.014) [82]. In contrast, they found no significant changes in improvement between before and during stimulation with different electrode locations (basal: 15 (4.5–29.5) % VAS-L reduction, *p* > 0.05; central: 25 (5–60) % VAS-L reduction, *p* > 0.05; apical: 4.5 (−6.5–39.5) % VAS-L reduction, *p* > 0.05; pitch-matched: 22.5 (2–28.5) % VAS-L reduction, *p* > 0.05). Furthermore, another study from Punte et al. (*n* = 7) reported a significant decrease in tinnitus loudness (VAS-L) after one week of stimulation through CI when all electrodes were activated (pre-implantation: 8.3 (SD: 1.1); 1 week: 4.4 (SD: 1.3), *p* = 0.027; 6 months: 3.5 (SD: 1.7), *p* = 0.042) [65]. The same study measured tinnitus loudness without providing stimulation and found that tinnitus loudness relapsed to its initial level. Rothholtz et al. (*n* = 1) showed a decrease in VAS-L during stimulation, but this decrease was not statistically tested (pre-stimulation: 5; during stimulation: 0; after stimulation: 6) [71]. Tinnitus loudness always increased after less than one second post stimulation.

In two studies, Mahmoudian et al. assessed the effect of promontory stimulation and reported the VAS-L of three groups: patients experiencing residual inhibition, patients without residual inhibition, and the control group. They showed a statistically significant decrease in VAS-L in the residual inhibition group and no statistically significant decrease in the non-residual group and the control group (Table 5b). A comparison between groups (*n* = 28) showed that the mean VAS-L of the residual inhibition group (*n* = 13) was significantly different from the non-residual inhibition (*n* = 15) and the control group (*n* = 28) (VAS-L residual inhibition group: 2.92 (SD: 1.75); a) compared to the VAS-L non-residual inhibition group: 6.73 (SD: 1.03), *p* < 0.05; b) compared to the VAS-L control group: 6.75 (SD: 1.23), *p* < 0.05) [62]. There were no statistically significant differences between the non-residual inhibition group and the control group. In a second study (*n* = 44), they showed the same findings (VAS-L residual inhibition group (*n* = 24): 3.13 (SD: 1.65); (a) compared to the VAS-L non-residual inhibition group (*n* = 20): 6.65 (SD: 1.04), *p* < 0.05; (b) compared to the VAS-L control group: 6.68 (SD: 1.25), *p* < 0.05) [61]. Wenzel et al. (*n* = 3) showed a decrease in VAS-L between activation of the round window implant and nine months but did not report a *p*-value (VAS-L activation: 8.0 (SD: 2.65), VAS-L nine months: 6.33 (SD: 5.51)) [69].

##### VAS-A

Wenzel et al. (*n* = 3) showed a decrease in VAS-A between activation of the round window implant and nine months but did not report a *p*-value (VAS-A activation: 8.67 (SD: 1.53), VAS-A 9 months: 6.33 (SD: 5.51)) [69].

#### 3.4.2. Parameters

The parameters tested can be found in Table 6. Due to the wide variety of devices, the stimulation patterns were not always described using the four operating parameters: electrode location, current level, pulse rate, polarity.

##### Electrode Location

Twelve studies directly compared the effect of different stimulated electrode locations (apical, middle, basal, or pitch-matched) on the experienced tinnitus. Arts et al. individually matched electrodes to the patient’s tinnitus pitch in 11 CI users [82]. They found that apical and central stimulation were more effective in terms of tinnitus loudness reduction, assessed by the VAS-L, than pitch-matched electrode stimulation (apical: 39% subjects with VAS-L reduction of 30% or more, pitch-matched: 22% subjects with VAS-L reduction of 30% or more (apical vs. pitch-matched: *p* = 0.042); central: 25 (IQR: 5–60) % VAS-L reduction, pitch-matched: 22.5 (IQR: 2–28.5) % VAS-L reduction (central vs. pitch-matched: *p* = 0.043)). Kloostra et al. (*n* = 19) reported that there was no statistically significant difference between single-electrode stimulation at apical electrodes, compared to basal electrodes for a reduction of at least one point in VAS-L (apical: 29%, basal: 19% stimulus conditions (*p* = 0.712)) [72]. Zeng et al. observed in their case study (*n* = 1) that total tinnitus suppression was achieved by stimulation of the four most apical electrodes, one by one, which could not be reached through stimulation of the most basal electrodes of the CI [70].

Several researchers performed electrical stimulation by placing the electrode on the round window or on the promontory and reported outcomes without statistical testing. Of these studies, entailing in total 84 patients, Aran and Cazals found that promontory stimulation resulted in self-reported total tinnitus suppression in 25% of patients and round window stimulation resulted in the same effect in 60% of patients [77]. Cazals et al. found self-reported total tinnitus suppression in 1 out of 6 patients using promontory stimulation and in 4 out of 6 using round window stimulation [75]. Portmann et al. described self-reported tinnitus reduction in 2 out of 7 patients using promontory stimulation and self-reported total tinnitus suppression in 4 out of 7 patients using round window stimulation [80].

##### Number of Activated Electrodes

Two studies using CI tested the effect of the number of activated electrodes on tinnitus loudness. Punte et al. (*n* = 14) showed that a statistically significant tinnitus loudness reduction occurred when all electrodes were activated, whereas activation of four or fewer basal electrodes did not provide significant tinnitus loudness reduction (Table 5) [65]. Kloostra et al. (*n* = 19) concluded that the effect of single-electrode stimulation on tinnitus was relatively insignificant in comparison to full-array stimulation and did not report statistical outcomes [72].

##### Current Level

Twenty-six studies assessed the effect of current level on tinnitus loudness. Kloostra et al. (*n* = 19) found statistically significantly greater tinnitus reduction, defined as a reduction of at least one point in VAS-L, using a moderate current level (C level) compared to near-threshold level (T level) (moderate current level: 30%, low current level: 18% stimulus conditions with a reduction of at least one point in VAS-L (*p* < 0.01)) [72]. Arts et al. (*n* = 10) found statistically significant differences between medium to loud stimulation through CI and a sham stimulation with no current provided (sham stimulation: 11 (IQR: −9–29) % VAS-L reduction; (a) compared to medium stimulation: 22.5 (IQR: 9.5–87.75) % VAS-L reduction (*p* = 0.033); b) compared to loud stimulation: 56.5 (IQR: −3.5–94) % VAS-L reduction (*p* = 0.014)) [82]. Chang et al. (*n* = 13) demonstrated that a current level eliciting a loud perception was significantly more effective in terms of VAS-L reduction than current levels eliciting soft perception (*p* = 0.027); further data were not reported in the publication [83]. Zeng et al. (*n* = 1) reported total tinnitus suppression at soft and comfortable levels [70].

##### Subthreshold vs. Suprathreshold Level

In the case study of Zeng et al. (*n* = 1), subthreshold stimulation through CI did not produce tinnitus suppression, whereas suprathreshold stimulation at low pulse rates did [70]. Hazell et al. (*n* = 9) found similar results using round window stimulation with AC and showed total suppression with a current level of about six dB more than the current level needed for hearing thresholds [76]. They did not perform statistical testing. In another study, Arts et al. (*n* = 11) found no statistically significant differences between subthreshold and suprathreshold electrical stimulation through CI (subthreshold: 7 (IQR: −4.5–29) % VAS-L reduction, soft level: 18 (IQR: −2.25–76) % VAS-L reduction, medium level: 22.5 (IQR: 9.5–87.75) % VAS-L reduction, loud level: 56.5 (IQR: −3.5–94) % VAS-L reduction (*p* > 0.05)) [82]. Additionally, Rothera et al. (16 ears) reported total tinnitus suppression at subthreshold level with anodic DC stimulation at the promontory window [66]. No statistical testing was performed because tinnitus was not the primary outcome of the study. This suppressive effect was only achieved at suprathreshold levels using AC.

##### Pulse Rate

Nineteen studies reported on the effect of stimulation with high or low pulse rates on tinnitus distress. Rubinstein et al. (*n* = 14) showed that high pulse rates (5000 pps) suppressed tinnitus in 45% (5/11) of patients with round window stimulation and in 33.3% (1/3) of patients with a CI [67] but did not perform statistical tests. Arts et al. (*n* = 10) did not find statistically significant differences between low (<2000 pps) and high (>2000 pps) rates (*p* = 0.81); no further data were provided [50,51]. Kloostra et al. (*n* = 19) found that tinnitus reduction was more often observed when stimulation through CI was at medium (26% of stimulus conditions) or high (29% of stimulus conditions) pulse rates, compared to stimulation at low (16% of stimulus conditions) pulse rates, but this effect was not significant (*p* = 0.493) [72]. Zeng et al. (*n* = 1) found that low rate (20–100 Hz) stimulation suppressed tinnitus through an apical electrode in a CI user [70].

Low pulse rates resulted in total tinnitus suppression in five studies investigating round window stimulation, of which three did not provide quantitative data [56] (*n* = 32), [58] (*n* = 11), [74] (*n* = 9). Okusa et al. (*n* = 65) identified low pulse rates as the most effective in tinnitus suppression (50 Hz: 40/65 total tinnitus suppression, 100 Hz: 33/65 total tinnitus suppression, 200 Hz: 20/65 total tinnitus suppression, 400 Hz: 16/65 total tinnitus suppression) [86]. Konopka et al. (*n* = 111) reported that better tinnitus reduction, measured as a reduction in tinnitus frequency of at least 1 kHz or in tinnitus loudness of at least 15 dB, was obtained using pulse rates below 1 kHz; however, this was not statistically significant [60].

Additionally, three studies reported on the importance of pulse rate in combination with other parameters. Chang et al. (*n* = 13) found a statistically significant interaction between pulse rate and current level (*p* = 0.03), in which the medium level was significantly more effective than soft (*p* = 0.043) or loud (*p* = 0.008) current levels, specifically at a high pulse rate (5000 pps); further data were not provided [83]. Similarly, Rothholtz et al. identified combinations of different pulse rates suppressing tinnitus in a single individual: a high rate (4638 pps) stimulus around the tinnitus pitch-matched electrode and a low rate (60 pps) stimulus at the most apical electrode of the CI [71]. According to Dauman et al. (*n* = 2), a low rate (125 Hz) enabled suppression of tinnitus at lower current levels compared with other rates (80, 250, 500 Hz); further data were not available [57].

##### Polarity

Six studies investigated the effect of polarity on tinnitus. Cazals et al. (*n* = 6) reported total tinnitus suppression in 5 out of 6 patients, only when the polarity of the direct current was anodic [75]. In another study, Aran and Cazals (*n* = 84) found that tinnitus improvement was achieved when an anodic direct current was applied [77]. Portmann et al. reported that changing the polarity from cathodic to anodic resulted in total tinnitus suppression in 14 out of 15 patients [80]. Rothera et al. showed self-reported tinnitus reduction in 1 out of 16 ears with anodic and cathodic current and in 5 out of 16 ears with the anodic direct current only [66]. Arts et al. (*n* = 10) tested anodic and cathodic first charged-balanced biphasic pulses, and 8 out 10 patients preferred the cathodic first charged-balanced stimulation as the most convenient configuration in terms of tinnitus loudness reduction [51]. In that study, anodic and cathodic pulses were tested but no significant difference in tinnitus reduction was found at the tinnitus pitch-matched electrode (*p* = 0.59), data were not available in the publication [50]. In the same manner, Péan et al. also asked four patients to choose between anodic or cathodic first charged-balanced oval window stimulation with regards to tinnitus severity; each of them opted for an anodic first pulse followed by a capacitive discharge [79].

#### 3.4.3. Harms

The status of harms reported can be found in Table 4 and are listed in Table 7. In the study of Olze et al., one out of four patients experienced an increase in tinnitus loudness, not during, but after intracochlear electrical stimulation [64]. In a case study, Zeng et al. reported the same observation after stopping the stimulation at low rate (100 Hz) and medium current level [70]. This effect was avoided by lowering the current level or introducing a gradual offset ramp in the waveform. No other harms were reported for intracochlear stimulation.

Two studies delivering extra-cochlear stimulation with high intensities (<300 μA) reported symptoms, such as dizziness and nausea, without providing the numbers of individuals affected [77], and an unpleasant effect in the head in one individual [78]. One study assessing promontory stimulation reported an increase in tinnitus severity in 12 out of 168 participants [59]. Among the 111 patients included in the study of Konopka et al., 4 patients experienced an increase in tinnitus loudness, 2 reported an increase in tinnitus frequency, and 1 mentioned an increase in tinnitus loudness after promontory stimulation [60]. In 2 out of 14 patients, promontory stimulation above approximately 400 μA evoked pain [67]. Ear drum perforation was reported in one out of four patients after implantation of a ball electrode placed on the promontory [63]. Graham et al. reported somatic sensations in all nine patients tested, an increase in tinnitus loudness in one patient and vertigo in four patients [74]. In the study of Okusa et al., 17 out of 65 patients reported dizziness, 5 reported discomfort of the throat, 3 reported discomfort of the nose, 1 developed a facial nerve palsy, and 1 had numbness of the face [86]. Other discomforts were reported in the study of Watanabe et al.: discomfort of the throat (*n* = 3/56), discomfort of the nose (*n* = 1/56), discomfort inside the mouth (*n* = 1/56) and discomfort of the lips and inside the mouth (*n* = 1/56) [68]. Wenzel et al. reported an increase in tinnitus loudness in the contralateral ear side of one out three patients, due to Meniere’s disease [69].

## 4. Discussion

So far, there are no therapies to directly counteract the origins of tinnitus, only tinnitus management therapies that reduce the burden. Since 1886 [88], attempts have been made to develop electrical stimulation patterns to suppress tinnitus. Tinnitus reduction has been reported as a positive effect of intracochlear electrical stimulation in studies on cochlear implantation in hearing-impaired people [89,90]. Some of these studies demonstrated therapeutic suppression of tinnitus symptoms, but there is no consensus on the most effective type of stimulus [7,8]. In this systematic review, we aimed to provide a comprehensible overview of the electrical intra- and extracochlear stimulation patterns studied and their effect on tinnitus.

The current study systematically reviewed the effect of intracochlear and extracochlear electrical stimulation for patients with tinnitus. A total of 89 patients out of 10 studies on intracochlear stimulation and 1109 patients out of 25 studies on extracochlear stimulation were included in this review. The included studies are heterogeneous in their methods, inclusion of participants, interventions and assessment of outcomes. There was a high to medium risk of bias in 22 out of 34 studies, especially due to lack of a non-exposed group and poor selection of the exposed group. All included studies showed subjective tinnitus improvement during or after electrical stimulation, using different stimulation patterns. Harms, including an increase in tinnitus loudness, were reported by 2 out of 89 patients tested with intracochlear stimulation and by 77 out of 1109 patients receiving extracochlear stimulation.

**Table 7 brainsci-11-01394-t007:** Harms reported in the included studies, with the number of participants in which harms were reported in brackets.

Author, Year	N	Configurations	Harms
Aran et al., 1981 [77]	84	PM, RW at current level <300 μA	dizziness, nausea
Cazals et al., 1978 [75]	6	PM	faint auditory sensations (2), tactile feelings (3)
Cazals et al., 1984 [78]	1	RW, (+), 5V	unpleasant effect in the head (1)
Graham et al., 1977 [74]	9	PM	increase in tinnitus loudness with a current level >5 mA at 100 Hz (1), somatic sensations (pain in the ear, numbness, vibration, tingling in the throat or cheek) (9), vertigo (4)
Konopka et al., 2001 [60]	111	PM	increase in tinnitus loudness (4), increase in tinnitus frequency (2), increase in tinnitus loudness after stimulation (1)
Konopka et al., 2008 [59]	168	PM	increase of tinnitus severity (12)
Matsushima et al., 1996a [63]	4	PM	ear drum perforation (1)
Olze et al., 2018 [64]	4	CI	increase in tinnitus loudness after stimulation (1)
Okusa et al., 1993 [86]	65	PM	dizziness (17), discomfort of the throat (5), discomfort of the nose (3), facial nerve palsy (1), numbness on the face (1)
Rubinstein et al., 2003 [67]	14	PM at 400 μA	pain (2)
Watanabe et al., 1997 [68]	56	PM	discomfort of the throat (3), discomfort of the nose (1), pain inside the mouth (1), cough (1), discomfort of the lips and inside the mouth (1)
Wenzel et al., 2015 [69]	3	RW	increase in tinnitus loudness in the contralateral side due to Meniere’s disease (1)
Zeng et al., 2011 [70]	1	CI at 100 Hz	increase in tinnitus loudness after stimulation (1)

A: amperes; CI: cochlear implant; N: numbers of patients reporting harms; PM: promontory stimulation; RW: round window stimulation; V: volts; (+): anodic polarity. The numbers in brackets correspond to the number of patients who reported the harm.

The evaluation of the effect of electrical stimulation was challenged by the heterogeneous patient selection in included studies. Study populations were highly heterogeneous in etiology of tinnitus, laterality of symptoms, duration of tinnitus and hearing profile, ranging from normal hearing to profound hearing loss. Tinnitus severity was not used as a selection criterion in all studies. A total of 19 out of 34 studies did not state their inclusion criteria based on tinnitus characteristics. Moderate or more severe tinnitus was an inclusion criterion in 13 studies. Inclusion based on tinnitus severity holds particular importance, as studies were designed specifically to measure treatment-related changes in tinnitus. However, given the data available, most patients presented at least moderate tinnitus distress before stimulation.

Self-reported tinnitus improvement was observed during or after electrical stimulation in each study. In studies that controlled for placebo effect, significant tinnitus reduction was reported only when electrical stimulation was applied [51,61,62,65]. This observation outlines the well-founded effect of electrical stimulation on tinnitus. However, the effect observed depends on the electrical patterns used and seems to be patient specific [82].

In a few cases, increase in tinnitus loudness, frequency and severity was reported during or after promontory stimulation [59,60,74] or stimulation through CI [64,70]. Other harms, such as vertigo, dizziness, and somatic sensations, were reported in few instances in studies investigating extracochlear stimulation [67,68,74,77,78,86]. These observations could be explained by the spread of electrical stimulation in the middle ear. The risk of developing harms related to electrical stimulation appears to be low. However, 19 out of 34 studies did not report harms in their methods or in the results. Therefore, the reporting of harms needs to be objectified in future studies.

To date, two reviews focused on electrical stimulation for tinnitus [7,25]. Zeng et al. identified opportunities and knowledge gaps in the use of electrical stimulation of the auditory nerve and the inner ear [7]. In this review, authors mentioned three different points of engagement: direct current stimulation, inner ear stimulation and auditory nerve stimulation. Zeng et al. suggested that the effectiveness of the different stimulation types depends on the etiology, the location and the type of tinnitus [7]. According to them, extracochlear stimulation is appropriate for patients with high-frequency tinnitus and normal audiograms. Another review focused solely on the effect of CI-programmed parameters for tinnitus [25]. Both reviews highlighted the differences between optimal stimulation parameters for speech perception and tinnitus suppression. Our study is the first to systematically review the effect of electrical stimulation of the inner ear for tinnitus relief, including intra- and extracochlear electrical stimulation.

We identified four main parameters characterizing stimulation patterns and having a potential influence on tinnitus: electrode location, current level, pulse rate and polarity. Some studies assessed a combination of parameters, whereas others aimed to evaluate the effect of a single parameter on tinnitus burden (Table A2). Most studies identified a combination of parameters effective in tinnitus suppression but were not able to isolate the effect of a single parameter on tinnitus. Moreover, the time of outcome assessment varied, ranging from during stimulation to days after stimulation. Given the aforementioned limitations, no comparison could be derived between the effect of intra- and extracochlear electrical stimulation. This heterogeneity in study design raises the question of what the best approach is to assess the effect of electrical patterns and, more specifically, the influence of each specific parameter. There is need for the establishment of a methodology to assess the effect of electrical stimulation patterns for tinnitus relief. In this context, a placebo condition or sham stimulation is essential in evaluating the effectiveness of electrical stimulation.

Apart from the consideration regarding methodology, authors assessing the effect of electrical stimulation should take special care to assess tinnitus changes in both ears. Notably, most studies included in this review reported tinnitus suppression in individuals but did not distinguish between the ipsilateral or contralateral ear. Among the ones who did observe this distinction, Portmann, Cazals, and Aran et al. reported that promontory or round window stimulation suppressed tinnitus only in the tested ear and had no effect on contralateral tinnitus or on tinnitus localized centrally [77,80,81]. However, other included studies showed that unilateral stimulation could improve tinnitus in the contralateral ear [38,76,85,87]. Thus, matching the tinnitus side and electrical stimulation location in the case of unilateral tinnitus as well as assessing tinnitus changes in both ears should also be considered in order to assess the effect of different stimulation strategies.

The underlying mechanisms of tinnitus and effect of electrical stimulation are not fully understood. Two main mechanisms might be involved in tinnitus suppression by electrical stimulation: a masking effect and a reduction effect [91]. The masking effect can be achieved using acoustic and electrical stimulation. The sound induced by electrical stimulation of the auditory nerve can reduce the contrast between the tinnitus signal and silence, which led to a decrease in tinnitus perception [92]. Nevertheless, researchers showed that inaudible stimulation can also suppress tinnitus in some patients [38,66,82,86]. This finding highlights another stimulation-based mechanism involved: the reduction mechanism, which modulates activity of the auditory cortex and suspends tinnitus generation. Aran and Cazals emphasized the dependence between the reduction effect and tinnitus origins [77]. They suggested location-specific management for tinnitus suppression. Based on their results, they hypothesized that electrical stimulation may only be effective if tinnitus originates at the periphery of the auditory pathway, whereas tinnitus of a more central origin cannot be improved by electrical stimulation. Some authors linked mechanisms underlying tinnitus to the effects of specific electrical patterns. For instance, Rubinstein et al. supported a theory of the deafferentation and alteration of normal spontaneous activity as the principal causes of tinnitus. Therefore, high pulse rate stimulation might produce spontaneous-like patterns, restoring abnormal activity and suppressing tinnitus percepts [67]. Rubinstein et al. only investigated high pulse rates and did not report comparison to other rates [67]. On the other hand, Zeng et al. used a case study to suggest that a low pulse rate might induce more robust and central activity, which would interfere with tinnitus-induced abnormal central activity [70]. Several questions remain regarding the extent to which a masking or reduction effect contributes to tinnitus suppression during and after electrical stimulation. Mallen et al. proposed a new audiological sequence (TILT) to isolate the masking and reduction effects of electrical stimulation [35]. Recent studies performed electrophysiological measurements, such as electrocochleography (ECochG) or auditory evoked potentials (AEPs), to further investigate the changes in neural activity induced by electrical stimulation (Table A3) [61,62,70]. Using objective measures, such as neuroimaging or electrophysiology, might be of additional value to better understand and optimize the effect of electrical stimulation on tinnitus.

The difference in reporting outcome can be of importance to the effect of electrical stimulation. The oldest studies did not have access to tinnitus questionnaires and assessed tinnitus changes based on self-reports [55,63,66,68,73,74,75,76,77,78,80,81,85,87]. More recent studies measured tinnitus severity using single-item questionnaires, multi-item questionnaires, or both. Among the 34 studies included, 10 studies monitored tinnitus changes, using tinnitus validated questionnaires. The change in tinnitus distress, burden or severity was often reported without introducing the notion of clinically relevant change. Self-reported tinnitus changes were available in almost every study included. However, the subjective changes reported do not belong to the same definitions and therefore, cannot be compared between studies. In the same way, comparing stimulation type is difficult as, for instance, DC stimulation cannot be translated to AC stimulation [65]. Considering the aforementioned limitations, no strong conclusion could be drawn from these data on differences between electrical stimulation patterns. This stresses the need for studies with adequate study designs and consistent selection of patients to provide homogeneity in outcomes.

Electrical stimulation has the potential to reduce tinnitus symptoms and has drawn attention in research for many years. Intracochlear stimulation through a CI is already highly developed for speech recognition in deafened patients. This technology could combine electrical stimulation to optimize both speech recognition and tinnitus suppression [7,8]. Nevertheless, due to its invasive approach, cochlear implantation can induce residual hearing deterioration in the ear implanted. Extracochlear stimulation for tinnitus can be provided using a basic pattern generator and has been extensively investigated, due to its minimal invasiveness. Whether this can be a perspective of tinnitus treatment in normal hearing patients remains to be seen. Moreover, little is known about the long-term effects of extracochlear electrical stimulation on tinnitus. Finally, significant challenges still need to be addressed on how to optimize electrical stimulation for maximum efficacy and whether tinnitus relief can be achieved without an auditory percept.

## 5. Conclusions

From the data included in this review, we concluded that electrical stimulation of the auditory nerve has potential for tinnitus suppression. Due to methodological limitations and the low reporting quality of the included studies, the potential of intra- and extracochlear stimulation has not been fully explored. To draw conclusions on which stimulation patterns should be optimized for tinnitus relief, a deeper understanding of the mechanisms involved in tinnitus suppression is needed, and new study designs should be considered. Further research is needed to determine the optimal electrical stimulation patterns to suppress tinnitus.

## Figures and Tables

**Figure 1 brainsci-11-01394-f001:**
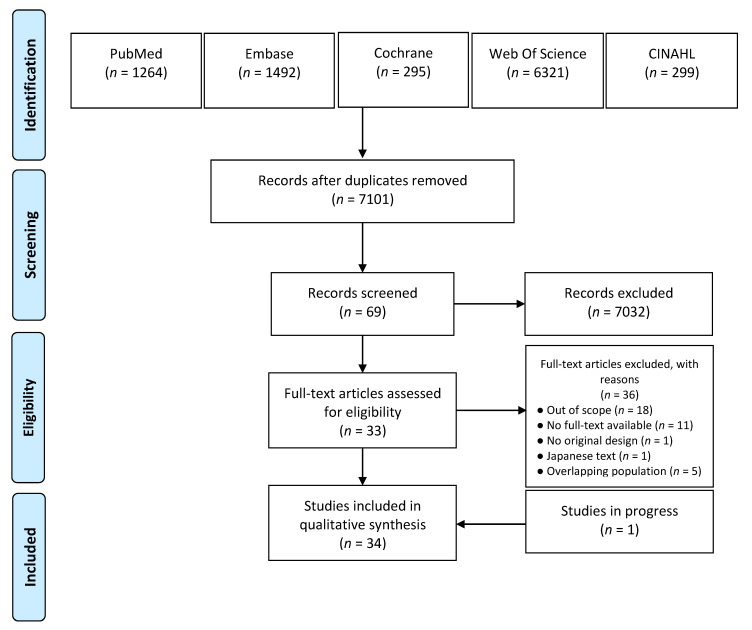
PRISMA flowchart of the literature search and study selection. Last date of search: 7 August 2021.

**Table 1 brainsci-11-01394-t001:** Search syntax (PubMed).

Search	Syntax
1	(((((((tinnitus[Title/Abstract]) OR ringing[Title/Abstract]) OR booming[Title/Abstract]) OR buzzing[Title/Abstract]) OR whizzing[Title/Abstract]) OR whistling[Title/Abstract]) OR blowing[Title/Abstract]) OR clicking[Title/Abstract] OR tinnitus[MeSH Terms])
2	(((((((((((electric*[Title/Abstract]) OR intracoch*[Title/Abstract]) OR extracoch*[Title/Abstract]) OR auditory[Title/Abstract]) OR experim*[Title/Abstract]) OR *cochle*[Title/Abstract]))) AND stim*[Title/Abstract])) OR electrical stimulation[MeSH Terms])
3	1 AND 2

**Table 2 brainsci-11-01394-t002:** Inclusion and exclusion criteria for the review.

	Inclusion	Exclusion
Participant	Adults (aged ≥18 years) with tinnitus	Studies focusing on children or animals
Interventions	Intra/extracochlear electrical stimulation to reduce tinnitus	Standardized CI stimulation patterns to rehabilitate hearing loss
Comparators	CI recipients with standard CI fitting or controls or no comparison groups	No exclusion restriction
Outcomes	Self-reported results of questions or questionnaires related to the experienced tinnitus	No self-reported measure or not related to the experienced tinnitus
Study designs	Case reports, cohort and randomized controlled trials	No original design, reviews, conference papers. No studies presenting overlapping population

**Table 6 brainsci-11-01394-t006:** Parameters assessed by the included studies. (**a**) Studies reporting on intracochlear electrical stimulation. (**b**) Studies reporting on extracochlear electrical stimulation.

(a) Studies Reporting on Intracochlear Electrical Stimulation						
Authors, Year	Stimulation Type (CI)	AC/DC	Parameter(s) Tested	Value Tested	Parameter Comparison in Terms of Tinnitus Reduction	*p*-Value
Arts et al., 2015 [82]	CI	AC	C	sham stimulation, −10, 20, 50, 80% DR	50% > sham stimulation	**0.033**
					80% > sham stimulation	**0.014**
			E	Basal (x1, x3), central (x1, x3), apical (x1, x3), pitch-matched (x1, x3)	Apical > pitch-matched	**0.042**
					Central > pitch-matched	**0.043**
			A	Random, sine wave, fixed	No comparison performed	NA
Arts et al., 2016 [51]	CI	AC	C	2.5–12.1 nC	Low > high	NI
			E	1–all	No comparison performed	NA
			P	(+), (−) first charge-balanced	No comparison performed	NA
			PR	200–5000 pps/channel	High > low	NI
			Pulse width	60–88 μs	No comparison performed	NA
			Dependency of environmental sounds	Independent, dependent	Not statistically different	>0.05
Chang et al., 2012 [83]	CI	AC	C	Soft, medium, loud	Loud > soft	**0.027**
					High rate: medium > soft	**0.043**
					High rate: medium > loud	**0.008**
			E	Apical, middle, basal	Not statistically different	NI
			PR	100–200, 5000 pps	Not statistically different	NI
Dauman et al., 1993 [57]	CI	AC	C	0.1–1.7 mA	No comparison performed	NA
			E	Apical, middle, basal	No comparison performed	NA
			PR	80, 125, 250 Hz	No comparison performed	NA
Kloostra et al., 2020 [72]	CI	AC	C	Near-threshold (T level), moderate (C level)	Moderate > near-threshold	**<0.001**
			E	Basal, apical Single electrode, full array	Not statistically different Not statistically different	0.712 NI
			PR	720, 1200, 2400 Hz	Not statistically different	0.493
Olze et al., 2018 [64]	CI	AC	A	Square wave	No comparison performed	NA
			PR	62 Hz	No comparison performed	NA
Punte et al., 2013 [65]	CI	AC	E	1, 2, 3, 4 most basal electrodes, all	All > most basal electrodes	**0.042**
Rothholtz et al., 2019 [71]	CI	AC	C	0–120 μA	No comparison performed	NA
			E	E1–E16	60 pps at most apical, high rates close to tinnitus matched pitch	NST
			PR	40–10000 Hz		
Rubinstein et al., 2003 [71]	CI	AC	C	300 μA–1.5 mA	No comparison performed	NA
			PR	4800 pps	No comparison performed	NA
			Pulse duration	25, 50, 80 μs/phase	No comparison performed	NA
			E	tinnitus pitch-matched	Not comparison performed	NA
Zeng et al., 2011 [70]	CI	AC	C	subthreshold, 0–10 loudness scale	1–6 > 7–10	NST
			E	Apical, basal	4 most apical > 4 most basal	NST
			PR	20–100, 5000 Hz	20–100 Hz > 5000 Hz	NST
**(b) Studies Reporting on Extracochlear Electrical Stimulation**						
**Authors, Year**	**Stimulation Type (RW, PM, OW)**	**AC/DC**	**Parameter(s) Tested**	**Value Tested**	**Parameter Comparison in Terms of Tinnitus Reduction**	* **p** * **-Value**
Péan et al., 2010 [79]	OW	AC	C	NI	No comparison performed	NA
			P	(+), (−) first charge-balanced	(+) > (−)	NST
			PR	NI	No comparison performed	NA
			Pulse shape	Square pulse, capacitive discharge	Square pulse followed by a capacitive discharge	NST
Daneshi et al., 2005 [56]	PM	AC	C	60–500 μA	No comparison performed	NA
			PR	50–600 Hz	No comparison performed	NA
Di Nardo et al., 2009 [58]	PM	DC+	C	0–500 μA	No comparison performed	NA
			PR	50–1600 Hz	50, 100 Hz > 200–1600 Hz	NST
Graham et al., 1977 [74]	PM	AC	C	1–100 μA	No comparison performed	NA
			PR	1–10000 Hz	No comparison performed	NA
Ito et al., 1994 [87]	PM	NI	NI	NI	No comparison performed	NA
Konopka et al., 2001 [60]	PM	DC+	C	20–600 mA	No statistically different	NI
			PR	60–10000 Hz	No statistically different	NI
Konopka et al., 2008 [59]	PM	AC	C	0.15–1.15 mA	No comparison performed	NA
			PR	Tinnitus pitch-matched	No comparison performed	NA
Mahmoudian et al., 2013 [62]	PM	AC	C	60–500 μA	No statistical difference between RI and NRI	0.61
			PR	1 Hz	No comparison performed	NA
			Frequency modulation	50 Hz	No comparison performed	NA
Mahmoudian et al., 2015 [61]	PM	AC	C	50–500 μA	No statistical difference between RI and NRI	>0.05
			PR	1 Hz	No comparison performed	NA
			Frequency modulation	50 Hz	No comparison performed	NA
Matsushima et al., 1994 [85]	PM	AC	A	Sinusoidal, 1 kHz	No comparison performed	NA
			C	0–70 μA	No comparison performed	NA
			PR	10 kHz	No comparison performed	NA
Matsushima et al., 1996a [63]	PM	AC	A	Sinusoidal, 100 Hz	No comparison performed	NA
			C	0–300 μA	No comparison performed	NA
			PR	10 kHz	No comparison performed	NA
Matsushima et al., 1996b [55]	PM	AC	A	Sinusoidal	No comparison performed	NA
			C	200 μA	No comparison performed	NA
			PR	10 kHz	No comparison performed	NA
Okusa et al., 1993 [86]	PM	AC	C	0–100 μA	No comparison performed	NA
			PR	50, 100, 200, 400 Hz	50 > 100 >200 > 400 Hz	NST
Rothera et al., 1986 [66]	PM	AC, DC	C	0–100 μA	AC suprathreshold, DC (+) subthreshold	NST
		DC	P	(+)/(−)	(+) > (−)	NST
		AC	PR	30–3000 Hz	No comparison performed	NA
Watanabe et al., 1997 [68]	PM	AC	C	5–160 μA	No comparison performed	NA
			PR	400 Hz	No comparison performed	NA
Carlson et al., 2020 [84]	PM	AC	C	NI	NA	NA
			E			
			P			
			PR			
Aran et al., 1981 [77]	RW, PM	AC, DC	Tinnitus side	Ipsilateral, contralateral	Ipsilateral > contralateral	NST
		AC, DC	E	RM, PM	RW > PM	NST
		AC	PR	Low (<100 Hz), high (>200 Hz)	High > low	NST
		AC	C	5–300 μA	No comparison performed	NA
	RW	AC, DC	P	(+)/(−)	(+) > (−)	NST
Cazals et al., 1978 [75]	RW, PM	AC, DC	C	20–300 μA	No comparison performed	NA
		AC, DC	E	RW, PM	RW > PM	NST
		AC, DC	P	(+)/(−)	(+) > (−)	NST
		AC	PR	>50–200 Hz	No comparison performed	NA
Cazals et al., 1984 [78]	RW	DC+	C	2, 5V	No comparison performed	NA
Hazell et al., 1993 [76]	RW	AC	A	Square, ramp, sinusoid	Sinusoid > (ramp, square)	NST
			C	0–300 μA	>+6 μA for hearing threshold	NST
			PR	10–200 Hz	(20–50 Hz) < 100 Hz	NST
House et al., 1984 [73]	PM	AC	A	Carrier wave dependent of sound	No comparison performed	NA
	RW (control)	AC	PR	60, 1600 Hz	No comparison performed	NA
Portmann et al., 1979 [80]	RW, PM	AC, DC	C	0–500 μA	No comparison performed	NA
			E	RW, PM	RW > PM	NST
			P	(+)/(−)	(+) > (−)	NST
			PR	50–6400 Hz	No comparison performed	NA
Portmann et al., 1983 [81]	RW, PM	DC+	C	1–5V	No comparison performed	NA
			E	RW, PM	RW > PM	NST
Rubinstein et al., 2003 [67]	RW	AC	C	300 μA–1.5 mA	No comparison performed	NA
			PR	4800 pps	No comparison performed	NA
			Pulse duration	25, 50, 80 μs/phase	No comparison performed	NA
Wenzel et al., 2015 [69]	RW	AC	C	0–3 mA	No comparison performed	NA
			PR	0–100 Hz	No comparison performed	NA
			Pulse duration	50 μs–8 ms	No comparison performed	NA

A: amplitude modulation; AC: alternative current; C: current level; CI: cochlear implant; DC: direct current; E: electrode location; NA: not applicable; NI: no information; NRI: non-residual inhibition group; NST: no statistical test performed; OW: oval window; P: polarity; PM: promontory; PR: pulse rate; RI: residual inhibition group; RW: round window. The *p*-value is the result of a statistical comparison test between the tinnitus questionnaire scores used for specific parameter values. Significant *p*-values are in bold.

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
