# Peer review of "Systematic Review on Intra- and Extracochlear Electrical Stimulation for Tinnitus"

_brainsci, 2021, doi:10.3390/brainsci11111394_

Round 1

Reviewer 1 Report

Review of:  Systematic review on intra- and extracochlear electrical stimulation for tinnitus

The authors perform a systematic review of intra and extra cochlear electrical stimulation for treatment of tinnitus.

The paper is very well written and I don’t see any serious issues with grammar.

With respect to the specific sections:

  1. Abstract – Brief and to the point. No changes recommended.
  2. Introduction is concise and discusses the issues to be addressed
  3. Methods section appropriately describes the method used for the systematic review including study selection, test for bias etc and utilizing the PRISMA guidelines for systematic review. Outcomes reviewed were self -reported questionnaire and adverse events. P values are mentioned but it’s not clear what statistical analysis was used. Also, how the heterogeneity was measured should be mentioned (I2analysis?)
  4. Discussion is to the point and indicates where the study adds to the literature. Adequate discussion of limitations of the study is made.

Overall, a very nice study and worthy of publication.  However more information on statistics used for the analysis is required.  Overall, this adds to the literature my performing a systematic review of relevant literature on electrical stimulation for treatment of tinnitus in humans.

I would recommend publication after addition of the information regarding statistical analysis.

  • I suggest adding information about statistical analysis

Author Response

Dear reviewer,

First of all, we want to thank you for your detailed feedback and review of our manuscript. We addressed your comment and feel we have improved our manuscript. We respond below to each point you raised.

  • P values are mentioned but it’s not clear what statistical analysis was used.

We agree with your valuable remark on the statistical analysis used. We added a sentence in the method section and stated more clearly in our revised manuscript that the p-values reported are the results of a comparison test between the tinnitus questionnaire scores (line 117-122): ‘The p-value is the result of a statistical comparison test between the tinnitus questionnaire scores used at different follow-up period or groups (Table 5) or for specific parameter values (Table 6). The cut-off of the p-value used to indicate a statistically significant result was established as described in the corresponding studies. We did not perform statistical analysis on the extracted data.’. The data compared in each study are detailed in the legend of the Table 5 and 6 (line 365-368, 375-377).

  • Also, how the heterogeneity was measured should be mentioned (I2analysis?)

We agree with your comments about the heterogeneity of the included studies and therefore decided to not include the meta-analysis in our systematic review. We did not perform I2 analysis because the included studies showed already heterogeneity in the inclusion of participants (no to severe hearing loss), the methods used (different interventions and comparators) and the assessment of outcomes (self-report to different tinnitus questionnaires), which were already limitations of performing a meta-analysis in this study (Cochrane Handbook 9.1.4). You can find this information in a paragraph of the method section (line 121-123): ‘Because of the heterogeneity of studies in methods, inclusion of participants, interventions and assessment of outcomes, we did not conduct a meta-analysis but instead performed a descriptive synthesis of the results.’

We look forward to your response. We would be glad to respond to any further questions and comments that you may have.

Yours sincerely,

Reviewer 2 Report

This study is a systematically review of the literature on the effectiveness of intra- and extracochlear electrical stimulation techniques and patterns as a treatment option for patients with tinnitus.

The research is original and relevant.

The paper is easy to read, the text is clear and the main question is widely elaborated.

From the data included in this review the conclusions are that electrical stimulation of the auditory nerve has potential for tinnitus suppression but further research is needed to determine the optimal electrical stimulation patterns to suppress tinnitus.

High risk of bias (0-3); Medium risk of bias (4-6); Low risk of bias (7-9) : colors instead black (line 321)

Author Response

Dear reviewer,

First of all, we want to thank you for your valuable remark and review of our manuscript. We agree with your comment and changed the color in the text accordingly.

We look forward to your response. We would be glad to respond to any further questions and comments that you may have.

Yours sincerely,

Reviewer 3 Report

Aim of the study is to review electrical stimulation patterns that induce tinnitus relief. Because of heterogeneity of existing studies, the authors did not conduct a meta-analysis but a descriptive review. Primary outcome is self-reported tinnitus experience according to standardised questionnaires and VAS. Secondary outcome is adverse effects.

The study is timely and very valuable because current tinnitus management focusses on cognitive behavioral therapies whereas directly influencing the tinnitus is being largely neglected. This is despite the positive experiences of many CI patients which suggests, that electrical stimulation may effectively reduce the tinnitus perception.

Minor comments are:
- Please explain in more detail why full text of some studies was not available.
- Throughout the text, it should be made clear that intracochlear stimulation is via CI (e.g. L. 171), and which parts of the text refer to stimulation via CI (e.g. LL. 255-319; 370-383; 417; 403-477). This is not always clear to readers unfamiliar with CI terminology and CI literature.
- Introduce abbreviation 'CI' in L. 39 and use it throughout the text (e.g. L. 395).
-LL. 160: Description of VAS is imprecise, also introduce VAS-L and VAS-A in the text.
- LL. 300-314: If studies are described in detail, number of participants of that study is a valuable information. For instance, the study by Wenzel et al., had only 3 participants  which renders p-levels to be misleading.
- LL. 395, 403: Tinnitus distress is not the same as tinnitus loudness.
LL. 421, 422: 'No statistically...' better start sentence with 'Another study...'.
- Results on long-term effects are missing. At least some studies produced data on tinnitus reduction following longer periods of electrical stimulation.
- Tables: Legend of table 3 does not match information in the table, tables 4-6 are too complex.
- The risk of adverse effects appears to be not very high, but this should be discussed in more detail.

Author Response

Dear reviewer,

First of all, we want to thank you for your detailed feedback and review of our manuscript. We addressed your comment and feel we have improved our manuscript. We respond below to each point you raised.

  • Please explain in more detail why full text of some studies was not available.

We agree with your suggestion and we added a sentence on the reason why full-text of 11 studies were not available: ‘Due to lack of response from corresponding authors contacted, full text was not available for 11 studies (37,38,47,39–46).’. For your information, we previously searched for the full text in Google Scholar, various databases (PubMed, Embase, Web of Science, etc.) and in ResearchGate, and then contacted the corresponding authors to retrieve the full text articles.

  • Throughout the text, it should be made clear that intracochlear stimulation is via CI (e.g. L. 171), and which parts of the text refer to stimulation via CI (e.g. LL. 255-319; 370-383; 417; 403-477). This is not always clear to readers unfamiliar with CI terminology and CI literature.

We agree with your remark and changed the text accordingly.

  • Introduce abbreviation 'CI' in L. 39 and use it throughout the text (e.g. L. 395).

We agree with your remark and changed the text accordingly.

  • 160: Description of VAS is imprecise, also introduce VAS-L and VAS-A in the text.

We agree with your remark and changed the text accordingly. We added a more detailed description of VAS: ‘The VAS consists of a horizontal or vertical line anchored at both end by a verbal descriptor referring to the tinnitus characteristics. The tinnitus characteristic is scored from 0 (not at all) to 10 or to 100 (extremely). A single question asks the patient to tick the line on the point that best matches to his or her tinnitus characteristic’.

  • 300-314: If studies are described in detail, number of participants of that study is a valuable information. For instance, the study by Wenzel et al., had only 3 participants which renders p-levels to be misleading.

We agree with your remark and added the number of participants for all the studies mentioned in the results section.

  • 395, 403: Tinnitus distress is not the same as tinnitus loudness.

We agree with your remark and changed the text accordingly.

  • 421, 422: 'No statistically...' better start sentence with 'Another study...'.

We agree with your remark and changed the text accordingly.

  • Results on long-term effects are missing. At least some studies produced data on tinnitus reduction following longer periods of electrical stimulation.

We provided the outcomes available at the longest follow-up period in Table 5 and reported it in the result section.

  • Tables: Legend of table 3 does not match information in the table, tables 4-6 are too complex.

We agree with your remark on Table 3 and changed the legend to match the information in the table.

We understand that the Table 4-6 seem complex, but they gather valuable data that are described in the text. We would like to keep these tables as there are, because they provide all the information necessary to summarize and compare the outcomes of the included studies.

  • The risk of adverse effects appears to be not very high, but this should be discussed in more detail.

We agree with your remark and added a short paragraph on the discussion on adverse events reported (LL. 561-569): ‘In a few cases, increase in tinnitus loudness, frequency and severity was reported during or after promontory stimulation (59,60,75) or stimulation through CI (65,71). Other harms such as vertigo, dizziness, somatic sensations were reported in few instances in studies investigating extracochlear stimulation (68,69,75,78,79,87). These observations could be explained by the spread of electrical stimulation in the middle ear.  The risk of developing harms related to electrical stimulation appears to be low. Though, nineteen out of 34 studies did not report harms in their methods or in the results. Therefore, reporting of harms needs to be objectified in future studies.’

We look forward to your response. We would be glad to respond to any further questions and comments that you may have.

Yours sincerely,
